# Mesenchymal MACF1 Facilitates SMAD7 Nuclear Translocation to Drive Bone Formation

**DOI:** 10.3390/cells9030616

**Published:** 2020-03-04

**Authors:** Fan Zhao, Xiaoli Ma, Wuxia Qiu, Pai Wang, Ru Zhang, Zhihao Chen, Peihong Su, Yan Zhang, Dijie Li, Jianhua Ma, Chaofei Yang, Lei Chen, Chong Yin, Ye Tian, Lifang Hu, Yu Li, Ge Zhang, Xiaoyang Wu, Airong Qian

**Affiliations:** 1Laboratory for Bone Metabolism, Xi’an Key Laboratory of Special Medicine and Health Engineering, Key Laboratory for Space Biosciences and Biotechnology, Research Center for Special Medicine and Health Systems Engineering, NPU-UAB Joint Laboratory for Bone Metabolism, School of Life Sciences, Northwestern Polytechnical University, Xi’an 710072, China; sofan@mail.nwpu.edu.cn (F.Z.); xiaoli225@mail.nwpu.edu.cn (X.M.); qiuwuxia@mail.nwpu.edu.cn (W.Q.); narutotcl@mail.nwpu.edu.cn (P.W.); zhangru@mail.nwpu.edu.cn (R.Z.); chzhh@mail.nwpu.edu.cn (Z.C.); suph@mail.nwpu.edu.cn (P.S.); zhangyan0911@mail.nwpu.edu.cn (Y.Z.); lidijie@mail.nwpu.edu.cn (D.L.); majianhua@mail.nwpu.edu.cn (J.M.); yangchaofei@mail.nwpu.edu.cn (C.Y.); chenlei1994@mail.nwpu.edu.cn (L.C.); yinchong42@mwpu.edu.cn (C.Y.); tianye@nwpu.edu.cn (Y.T.); hulifang@nwpu.edu.cn (L.H.); liyu@nwpu.edu.cn (Y.L.); 2Law Sau Fai Institute for Advancing Translational Medicine in Bone and Joint Diseases, School of Chinese Medicine, Hong Kong Baptist University, Hong Kong 999077, SAR, China; zhangge@hkbu.edu.hk; 3Ben May Department for Cancer Research, The University of Chicago, Chicago, IL 60637, USA; xiaoyangwu@uchicago.edu

**Keywords:** MACF1, bone development, bone formation, osteoporosis, mesenchymal stem cells, osteogenic differentiation, SMAD7

## Abstract

Microtubule actin crosslinking factor 1 (MACF1) is a large crosslinker that contributes to cell integrity and cell differentiation. Recent studies show that MACF1 is involved in multiple cellular functions such as neuron development and epidermal migration, and is the molecular basis for many degenerative diseases. MACF1 is highly abundant in bones, especially in mesenchymal stem cells; however, its regulatory role is still less understood in bone formation and degenerative bone diseases. In this study, we found *MACF1* expression in mesenchymal stem cells (MSCs) of osteoporotic bone specimens was significantly lower. By conditional gene targeting to delete the mesenchymal *Macf1* gene in mice, we observed in MSCs decreased osteogenic differentiation capability. During early stage bone development, the MACF1 conditional knockout (cKO) mice exhibit significant ossification retardation in skull and hindlimb, and by adulthood, mesenchymal loss of MACF1 attenuated bone mass, bone microarchitecture, and bone formation capability significantly. Further, we showed that MACF1 interacts directly with SMAD family member 7 (SMAD7) and facilitates SMAD7 nuclear translocation to initiate downstream osteogenic pathways. Hopefully these findings will expand the biological scope of the *MACF1* gene, and provide an experimental basis for targeting MACF1 in degenerative bone diseases such as osteoporosis.

## 1. Introduction

Mesenchymal stem cells (MSCs) are important cell types responsible for bone formation. MSCs differentiate into a series of bone forming lineage cells that are involved in osteogenic differentiation and bone formation, and this process is subtly regulated. In the ageing process, impaired MSCs function and suppressed osteogenic differentiation are direct causes for bone diseases such as osteoporosis. However, the regulation of MSCs osteogenic differentiation is complicated, and the underlying mechanism remains to be further studied.

The microtubule actin crosslinking factor 1 (MACF1) is an important mammalian spectraplakin protein that interacts with the cytoskeleton [1,2,3]. As one of the largest mammalian proteins (≈620 KD), MACF1 has the ability to bind to actin filaments with its *N*-terminal calponin homology domain, and interact with microtubules with the *C*-terminal EF-hand/GAR domain [2,4] (Appendix A). MACF1 also has a homologous sequence similar to the Smc family of ATPases [4,5], and is reported to participate in MT-mediated cargo transport [5,6,7]. MACF1 is expressed ubiquitously as early as at E7.5 during the embryonic period [6] and plays essential roles in regulating various development processes in mammals [8,9,10]. With especially high abundance in the heart, lung, brain, and bones, its unique structure enables MACF1 a variety of functions in neuron development, epidermal migration, and more [11,12,13], analysis of this evidence shows that MACF1 is crucial for normal cells to proliferate, polarize, and differentiate. Reduction or depletion of MACF1 disorganizes the cytoskeleton and impairs cellular functions [12,14,15,16], and further confers risk for degenerative diseases such as inflammatory colitis and Parkinson’s disease [9,14,15,17].

MACF1 is highly abundant in bone tissue, especially in bone forming cells such as bone marrow mesenchymal stem cells (MSCs), however, its function in bone development regulation remains poorly understood. Previously we showed that femoral *Macf1* level is significantly lower in both osteoporotic mice [18] and aged mice [19], in vitro knockdown of MACF1 inhibits cell cycle and differentiation in preosteoblasts [18,19,20], and skull transfection of foreign MACF1 plasmid shows a promoting effect on osteoblast differentiation and bone formation [21]. These results indicate potential key functions of MACF1 in osteoblast differentiation and bone formation. However, the osteogenic functions of MACF1 in MSCs are still less understood, and no animal models are yet available to depict its in vivo roles in bone formation regulation. We wonder how mesenchymal MACF1 regulates bone development and formation. Moreover, as a giant structural protein, we wonder in particular what downstream factors are involved in MACF1-mediated bone formation regulation.

In this study, by gene manipulation to delete MACF1 in the mesenchyme in mice, we identified essential roles of MACF1 in regulating osteogenic differentiation and bone formation in vivo. We show that MACF1 is essential for early stage bone development, loss of MACF1 attenuates osteogenic potential in MSCs, and further retards bone development and impairs bone properties and bone strength. In addition, we show that MACF1 directly interacts with SMAD family member 7 (SMAD7) and facilitates SMAD7 nucleus translocation to initiate downstream osteogenic pathways. By coupling the results obtained, we have identified key roles of MACF1 in maintaining osteogenic differentiation and bone formation and advanced the potential for exploiting MACF1 as a new therapeutic target for degenerative bone diseases such as osteoporosis.

## 2. Materials and Methods

### 2.1. Gene Expression Profile

Gene expression profile data were downloaded from NCBI Gene Expression Omnibus (GEO, ncbi.nlm.nih.gov/geo/) database (GSE35959, GSE12274). These datasets contain expression profile data of mesenchymal stem cells (MSCs) from primary osteoporosis patients. Detailed descriptions are accessible through GEO accession number or original research papers [22,23]. Analysis was performed using the series_matrix file. For multiple probes of the same gene, the maximum value was used as gene expression value.

### 2.2. Human Bone Specimen

Human bone specimens were proximal femora collected from inpatients suffering femoral neck fracture, these patients fractured by accident, and needed operational internal fixation or femoral head replacement. Subjects included in the study were postmenopausal female aged at their 60s or 80s, those who suffered severe diseases or received osteoporosis treatment were excluded. Fresh bone specimens were rinsed with normal saline and preserved in RNAlater under −80 °C. Samples were collected in Hong Hui hospital (Xi’an, China), informed written consents were obtained from each subject before the experiment. The experiment was approved by Biomedical Research Ethics Committee of Hong Hui hospital (3 March 2017) and Institution Review Board of the Northwestern Polytechnical University (12 September 2016).

### 2.3. Generation of the MACF1 Conditional Knockout (cKO) Mouse

Mice carrying the floxed MACF1 alleles (in B6 background) were acquired through a material transfer agreement between the University of Chicago (USA) and the Northwestern Polytechnical University (China). Construction of the targeting vector was reported previously [5], in which the loxP sites were integrated to excise exons 11, 12, and 13 of the *Macf1* gene (while in the original literature, the flanked exons were reported as E6 and E7) (Appendix A). Mice expressing Cre recombinase under the control of Prx1 (paired related homeobox1) promoter (in B6 background) were acquired from Biocytogen Co., Ltd. (Beijing, China). The MACF1 conditional knockout (cKO) mice were generated by breeding the MACF1 Flox mice to the Prx1-Cre mice. Offspring genotype was determined by PCR analysis of genomic DNA from the toe tissue (or for embryos, from liver tissue).

Feed (Beijing KeaoXieli Feed Co., Ltd., Beijing, China, SPF mouse feed, ^60^Co irradiated) and water (boiled water) used for mice husbandry were pre-processed properly, and were all available ad libitum to mice. Breeding and number of mice were controlled according to the 3Rs principle. Mice used for planned experiments were euthanatized after overdose injection of pentobarbital sodium (i.p., 1 g/Kg), while others were euthanatized by CO_2_ inhalation. All mice experiments were performed according to the Guide for the Care and Use of Laboratory Animals and under the permission of Laboratory Animal Ethics and Welfare Committee of the Northwestern Polytechnical University (12 September 2016).

### 2.4. Study Design and Animal Grouping

To study the in vivo functions of MACF1, age-matched same-sex controls were set up (mice numbers are indicated in each figure). Unless otherwise stated, the MACF1^f/f^;Prx1^Cre/+^ mice (cKO) were considered to have the *Macf1* gene deleted specifically in MSCs, while the MACF1^f/f^ mice (Flox) were used as controls. All mice were maintained in barrier condition (SPF lab, 12/12 h light cycle, 24 ± 1 °C, 30–70% RH) with no more than four in each cage.

### 2.5. Cell Preparation

Mouse mesenchymal stem cells were isolated from compact bone of 6-week old male mice [24]. Briefly, mice femora were isolated, cleaned, and transferred to a 60 mm culture dish containing 5 mL α-MEM (supplemented with 0.1% penicillin/streptomycin and 2% FBS). Epiphyses in both ends were cut open for complete removal of bone marrow. Bone shafts were then excised into chippings and digested (α-MEM + 10% FBS + 1 mg/mL Type II collagenase, 37 °C, 200 rpm, 1 h), bone chips were then collected and seeded in a 60 mm dish for culture (37 °C, 5% CO_2_). Culture medium was changed on the third day to remove loosely attached cells and tissue debris. Cells at passage 2 to 5 were used for experiments.

Construction of the MACF1 knockdown MC3T3-E1 cell line (KD), and transfection of pKH3S-ACF plasmid into MACF1 KD cells were reported previously [20,21].

Growth medium (10% FBS + 1% pen-strep + 1% L-glutamine in α-MEM) were used for subculture, and osteo-induction medium (10% FBS + 1% pen-strep + 1% L-glutamine + 0.1 μM dexamethasone + 50 μg/mL ascorbic acid + 10 mM beta-glycerophosphate in α-MEM) were used for osteo-induction.

### 2.6. RNA and Real Time Quantitative PCR (qPCR)

High quality total RNA from cell and tissue samples were isolated using TRIzol™ reagent according to the manufacturer’s instructions. RNA quality was determined by ultraviolet spectrophotometry. cDNA was synthesized on 1 μg total RNA using the cDNA synthesis kit (TaKaRa RR037A, Kusatsu, Japan) following the manufacturer’s instructions. Real-time quantitative PCR was performed using the SYBR^®^ Green method on 2 μL cDNA (1:50) in a 20 μL PCR system. PCR data were analyzed with the comparative CT method (2^−ΔΔ*C*^_T_). *Gapdh* or 18s was used as internal controls.

### 2.7. Protein and Western Blot

For extraction of protein from bones, femora were excised free of connecting tissue, flash-frozen, and pulverized into bone chips using a pestle and mortar in the presence of liquid nitrogen (while soft tissues were pulverized using a tissue pulverizer (PRIMA PB100-SP08, London, UK), and then digested with protein lysis buffer (Beyotime P0013, supplemented with 1× protease inhibitor cocktail). The lysate was then centrifuged at 12,000× *g*, 4 °C for 2 min, and the supernatant was collected. For fractionation of cytoplasmic and nuclear proteins, cell cultures were washed and lysed using the Nuclear and Cytoplasmic Protein Extraction Kit (Beyotime P0028, Shanghai, China) following the manufacturer’s instructions. Protein concentrations were determined using the BCA method.

For Western blot analysis, protein samples separated by SDS-PAGE were transferred onto PVDF membranes and probed with antibodies against target proteins. Briefly, 20 μg protein samples were electrophoresed at 100 V on 5% stacking gel/10% separating gel. After transferred to PVDF membrane (100 V, 90 min), and incubated with 5% non-fat milk for 30 min, the membrane was probed with primary antibodies against MACF1 (1:1000), SMAD7 (1:1500), Lamin B1 (1:2000), and GAPDH (1:2000). Washed membranes were then incubated with HRP-conjugated secondary antibodies, and developed and imaged (Gel Doc^™^ XR+ Gel System).

### 2.8. Immunofluorescence

Immunofluorescence staining was performed on fibronectin-coated coverslips. Briefly, cells were plated on coverslips (1 × 10^4^/cm^2^) and cultured under normal condition for 12 h. Then cells were fixed with 4% PFA and rinsed for 5 min with TBS, 10 min with TBS-0.5% TritonX-100 (TX), and 3 × 5 min with TBS-0.1%TX, and then blocked with 2% BSA for 10 min. Primary antibodies were added onto each coverslip (10 μL, 1:50), and incubated at 4 °C overnight. The next day, after being washed with TBS-0.1%TX, fluorescein-labeled secondary antibodies were added (10 μL, 1:100), and incubated for 1 h. Nucleuses were stained using 5 μg/mL DAPI (4′,6-diamidino-2-fenilindol) (236276-10mg, Shanghai Root Biotec, Shanghai, China). Mounted coverslips were then examined by confocal microscopy (Leica TCS SP8, Wetzlar, Germany).

### 2.9. Co-Immunoprecipitation and iTRAQ-LC-MS/MS

Co-immunoprecipitation was performed using freshly harvested MC3T3-E1 cells incubated with antibody recognizing the MACF1 protein. Briefly, cells were harvested in RIPA buffer. Whole cell lysates were incubated at 4 °C overnight with 4 µg anti-MACF1 antibody (Abcam, Cambridge, MA, USA, ab117418) or 2 µg Rabbit IgG (abcam, Cambridge, MA, USA, ab46540). The antigen/antibody complexes were then mixed with 40 µL recombinant Protein A+G agarose beads (Beyotime P2055, Shanghai, China), and incubated for 2 h at room temperature. Immunocomplexes were then centrifuged at 1000× *g* for 5 min to remove supernatant, and then beads complexes were washed with RIPA buffer for 5 times. After washing, beads complexes were resuspended with 20 µL 1× SDS-PAGE electrophoresis loading buffer, and bound proteins were retrieved from agarose beads by boiling for 5 min.

For analysis of MACF1-interacting proteins, co-immunoprecipitated products were labeled with iTRAQ (Isobaric tags for relative and absolute quantification) reagents and subjected to liquid chromatography followed by tandem mass spectrometry (LC-MS/MS) analysis.

### 2.10. Cleared Skeleton Preparation

Fetuses or neonatal mice were euthanized by anesthetization (pentobarbital sodium, i.p., 0.5 g/kg). Samples were skinned and eviscerated before being fixed in 95% ethanol for 24 h, and incubated in acetone overnight. To stain for cartilage, samples were rinsed with water for three changes and left in 0.03% (*w*/*v*) Alcian blue solution for about 18 h, and then cleared in 1% KOH until soft tissues were hardly visible. To visualize calcified bone, samples were counterstained in 0.02% Alizarin Red S solution for about 12 h, and then cleared in 1% KOH. Images were acquired by a digital camera, and dimensional parameters were acquired using Image-Pro Plus (Media Cybernetics, Silver Spring, MD, USA).

### 2.11. Bone Densitometry and Micro-Computed Tomography

Bone mineral density was measured using a high resolution dual-energy X-ray imaging system (InAlyzer, MEDIKORS Inc., Seoul, Korea; 55 KeV and 80 Kev). Microarchitecture and stereological parameters were acquired using an micro-computed tomography instrument (EL-SP, GE, 80 kPV, 80 μA, 3000 ms) at an isotropic resolution of 8 μm. Data were then reconstructed at a voxel size of 16 × 16 × 16 μm. Stereological parameters of the trabeculae were measured within a 1 mm-high region above the distal femur epiphyseal plate using MicroView (a GE distribution, Chicago, IL, USA; advanced 3D ROI). Sample scanning and data analysis adhered to a previous protocol [25]. Additional parameters were acquired using ImageJ (NIH; Bethesda, MD, USA) and Image-Pro Plus (Media Cybernetics, Rockville, MD, USA).

### 2.12. Three-Point Bending Test

Mechanical properties of the femur were tested by means of three-point bending using a universal test machine (Instron 5943, Norwood, MA, USA). Briefly, freshly dissected femora were cleaned free of connective tissue and wrapped in normal saline-soaked gauze and kept moist at 4 °C. For subsequent tests, femora were secured to the lower supporting pins (span: 9 mm, fillet radius: 1 mm) one by one. After preloaded with 1 N for 5 s, the tissues were loaded to failure at 1.5 mm/min, and load-displacement curves were collected. The inner and outer diameters of the loaded femora were measured using a 3D digital microscope (Hirox KH-8700, Tokyo, Japan). Mechanical parameters were acquired using a built-in software and a previous reported method [26].

### 2.13. Histochemical, Calcein Labeling, and Histomorphometric Analysis

Formalin-fixed paraffin-embedded (FFPE) tissue sections were prepared and stained using conventional methods. Briefly, fresh tissue was fixed in 4% PFA, decalcified in 10% EDTA, dehydrated through graded ethanol, and embedded into paraffin. Sections were obtained at 4 μm. H&E, ALP (Alkaline phosphatase) staining (0.5% β-glycerophosphate Disodium + 0.5% Barbital sodium + 0.5% CaCl_2_ + 0.5% Mg_2_SO_4_ + 2% cobalt nitrate + 1% ammonium sulfide) and TRAP staining (50 mM C_4_H_12_KNaO_10_) were performed on serial sections. Stained sections were then scanned using a digital slide scanner (Aperio AT2, Leica, Wetzlar, Germany).

For plastic embedded sections, 6-week-old mice were injected intraperitoneally with calcein (10 mg/kg) 10 and 2 days prior to euthanasia, respectively. Femora were then collected and fixed, and embedded in gradient methyl methacrylate after conventional tissue dehydration and clearing. Sections (≈20 μm) were prepared using the EXAKT 300CP tissue cutting systems and stained using modified Goldner’s trichrome staining method [27]. Slides were visualized using a phase contrast microscope (BIOQUANT OSTEO-SCAN, Nashville, TN, USA). For histomorphometric parameters, images were analyzed using Histomorph suite [28] and Image-Pro Plus (Media Cybernetics, Rockville, MD, USA).

### 2.14. Serological Factors

Whole blood collected by cardiac puncture was used for serum separation. Serological factors were detected by means of enzyme-linked immunosorbent assay following the manufacturer’s instructions (Cloud-Clone Corp., Katy, TX, USA). O.D. values were read at 450 nm (BioTek Synergy Winooski, HT, VT, USA).

### 2.15. Cell Culture Staining

ALP staining was performed using a BCIP/NBT alkaline phosphatase color development kit (Beyotime C3206, Shanghai, China) following the manufacturer’s instructions. Briefly, cell cultures were washed with cold PBS, fully fixed in 4% PFA, and washed. The staining working solution was prepared by mixing BCIP (5-Bromo-4-chloro-3-indolyl phosphate, 300×) and NBT (nitro blue tetrazolium chloride, 150×) into the color developing buffer, cell cultures were then stained in the dark for 15 min. Nodules formation staining was performed using 0.5% Alizarin Red S solution (ARS). Von Kossa staining was performed using 5% AgNO_3_ after being induced for 21 days. Stained cell culture plates were scanned using a digital scanner (CanoScan 9000F MarkII, Tokyo, Japan).

### 2.16. Electric Cell-Substrate Impedance Sensing

Dynamic physiological properties of MSCs during proliferation and differentiation were monitored non-invasively using the ECIS (electric cell-substrate impedance sensing) Zθ system (Applied BioPhysics, Troy, NY, USA) [29]. Briefly, cells plated onto fibronectin-coated (20 μg/mL) standard 8-well array (8W10E+) were supplemented with 400 μL growth medium, and cultured in CO_2_ incubator for 6 h before the initial test. To measure impedance values of differentiating MSCs, growth medium was replaced with osteogenic osteo-induction medium, and data were collected every 80 s at a current frequency of 64 kHz.

### 2.17. Plasmid Preparation

The SMAD7 CDS (coding sequence) region was cloned into the pcDNA3 backbone for construction of the SMAD7 overexpression plasmid. Briefly, double enzyme digestion was performed on selected restriction enzyme sites. Then, seamless clone-specific primers were designed to match selected restriction enzyme sites, and the SMAD7 CDS sequence was amplified by PCR. Products from the above steps were purified by gel extraction. Linearized vector and CDS insert were recombined in the presence of high-fidelity polymerase. The recombinants were then transformed into Stbl3 competent cells and sequenced. Endotoxin-free plasmid was prepared and transfected into cells using nano-particle transfection reagent (Engreen H4000, Beijing, China) according to the manufacturer’s instructions.

### 2.18. Data Processing and Statistics

Image acquisition equipment included phase contrast microscope, fluorescence microscope, confocal microscope, digital camera, digital scanner, digital slide scanner, gel imaging system, micro-computed tomography, and dual-energy X-ray absorptiometry. To ease observation and statistical analysis, certain images may have been pre-processed using Adobe Photoshop^®^ (Adobe Inc., San Jose, CA, USA). Operations applied to image data included cropping, scaling, flipping, background clearing, local zoom in, and tagging. Other lossy operations such as brightness adjustment, contrast adjustment, levels adjustment were applied only to a limited number of images simultaneously, and great efforts were made to ensure that these operations did not change the essence or any measurable variables of the images.

All data are presented as mean ± s.d. Statistical significance between two groups was determined using Student’s *t*-test. A *p*-value less than 0.05 was considered statistically significant.

## 3. Results

### 3.1. Mesenchymal Microtubule Actin Crosslinking Factor 1 (MACF1) Is Decreased in Osteoporosis Patients

MACF1 is a large structural protein [2,30]. To verify the involvement of MACF1 in primary osteoporosis, we firstly analyzed the gene expression profile in human MSCs of elderly osteoporotic patients. In a dataset (GSE35959) containing samples from middle-aged, aged but non-osteoporotic, and aged osteoporotic patients (Appendix A), we found decreased level of MACF1 in the aged osteoporotic group (Appendix A). Further analysis revealed that multiple osteogenic-related genes were negatively correlated with the degrees of osteoporosis (Appendix A), and MACF1 was positively correlated with these osteogenic-related genes (Appendix A). Similar results were obtained in another dataset (GSE12274) containing samples from young, middle-aged, and aged osteoporotic patients (Appendix A). These data suggest that mesenchymal MACF1 is significantly downregulated in osteoporotic patients. Next, MSCs were isolated from bone specimens of aged osteoporotic patients, we found that the expression of either MACF1 or osteogenic-related genes in the 80-year-old group were significantly lower compared with the 60-year-old control group (Figure 1A). Lastly, aged osteoporotic mice were examined, with increased age and continuously decreased bone mass (Appendix A), and the expression of the *Macf1* gene decreased accordingly (Figure 1B). These data suggest that MACF1 is closely related to primary osteoporosis.

### 3.2. Mesenchymal Deletion of Microtubule Actin Crosslinking Factor 1 (MACF1)

Previous studies show that conventional inactivation of the *Macf1* gene leads to teratogenicity and lethality during development in mice [6], suggesting MACF1 is crucial for embryo development. In order to study the in vivo role of MACF1 in bone development and formation, we utilized the Cre/Loxp technology and constructed a genetically modified mouse model in which the *Macf1* gene was deleted specifically in mesenchymal stem cells (Flox, Macf1^f/f^; cKO, Macf1^f/f^;Prx1^Cre/+^) (Appendix A). Similar to those conditional MACF1 deficient mice generated using other Cre lines, mesenchymal deletion of MACF1 did not significantly affect body length or body weight in neonates (Figure 1C) and Appendix A) [31], and offspring were obtained in an expected sex ratio and Mendelian frequency (Appendix A). By adulthood, the difference of body size and organ coefficient in cKO and Flox groups were not significant as well (Appendix A), suggesting that mesenchymal deletion of MACF1 did not apparently change fundamental physiological indications in mice.

We then detected MACF1 expression levels to confirm successful deletion of MACF1. Real-time quantitative PCR (qPCR) analysis showed that in the cKO group, mRNA level of *Macf1* was significantly lower in bone tissue, but not in other tissues like muscle, cerebrum, or liver (Figure 1D). It is worth noting that the *Macf1* gene in cKO was also lower in the brown adipose tissue (Figure 1D). Then, immunoblotting and immunofluorescence experiments further demonstrated that MACF1 protein was barely detectable in MACF1 cKO MSCs (Figure 1E). These results indicate that conditional targeting of MACF1 was successful, and that mesenchymal deletion of MACF1 does not change fundamental physiological indications in mice.

### 3.3. Loss of Microtubule Actin Crosslinking Factor 1 (MACF1) Inhibits Osteogenic Differentiation in Mesenchymal Stem Cells (MSCs)

Although the in vitro roles of MACF1 have been studied in cells such as neuron and epidermis cells, much is still unknown in bone forming cells. To verify whether mesenchymal deletion of MACF1 generated phenotypes, and to study the osteogenic functions of MACF1, bone derived MSCs from the MACF1 cKO mice were isolated and cultured. Alkaline phosphatase (ALP) and Alizarin Red S (ARS) staining showed that during osteo-induction, MSCs lack of MACF1 had less ALP activity and less mineralized nodule formed (Figure 1F,G), von Kossa staining further confirmed that 21-days after osteo-induced culture, and cKO cells showed less mineralization capability (Figure 1H).

Next, proliferation and differentiation capability of osteo-induced MSCs were detected non-invasively by electric cell-substrate impedance sensing (ECIS). By monitoring real-time impedance values during osteo-induction, the ECIS curve showed that during exponential growth, proliferation capability decreased significantly in MACF1 deficient MSCs (Figure 1I, left), population doubling time increased from 39.8 h in Flox cells to 50.5 h in cKO cells. In addition, loss of MACF1 significantly suppressed differentiation capability of MSCs, especially in the early phase of osteogenic differentiation (Figure 1I, right).

Lastly, qPCR analysis showed that, during osteogenic differentiation, mRNA levels of either Macf1 or osteogenic genes decreased in cKO cells (Figure 1J and Appendix A), suggesting impaired osteogenic differentiation capability upon MACF1 deletion. These data indicate that MACF1 is necessary for maintaining osteogenic differentiation in MSCs, and that lack of MACF1 impairs osteogenic functions in MSCs.

### 3.4. Mesenchymal Deletion of Microtubule Actin Crosslinking Factor 1 (MACF1) Impairs Early-Stage Bone Development in Mice

Previous studies show that Prx1-driven Cre recombinase is expressed in a subset of craniofacial mesenchyme, and in early limb bud mesenchyme [32,33]. To study the impact of MACF1 on early stage bone development, we first examined the skull at late embryonic stage as well as in neonatal mice. Cleared skeletal preparation of the skull showed obvious delays in suture fusion and calvarial ossification in cKO group (Figure 2A and Appendix A). MicroCT experiment then substantiated the retardation in suture closure, and showed that parietal bone was thinner in cKO mice at P1 (Figure 2B). In addition, formalin-fixed paraffin embedded (FFPE) skull sections were also prepared. HE staining confirmed attenuation in calvaria thickness, while alkaline phosphatase (ALP) staining showed that cKO mice had less ALP expression in parietal bone (Figure 2C), however Alcian blue staining of the skull did not show a significant difference between Flox and cKO mice (Appendix A).

Unlike flat bones, bones of the trunk and extremities are formed with an initial hyaline cartilage. To further investigate the effect of MACF1, long bones were also scrutinized. Cleared bone preparation revealed that during early stage bone development, mineralized bone formation in femur and tibia was slower in cKO mice (Figure 2D). Taken together, these results suggest that MACF1 is crucial for endochondral ossification and intramembranous ossification, and loss of MACF1 retards early-stage development and formation of the skeleton in mice.

### 3.5. Microtubule Actin Crosslinking Factor 1 (MACF1) Is Required for Bone Formation

Prx1-driven Cre recombinase is also expressed in extremities in mammals [34,35]. In adults, bone forming cells derived from mesenchymal stem cells are responsible for bone matrix synthesis and mineralization [36], and finally form the mineral structure to support bone functions. To investigate the roles of MACF1 on bone phenotype in adults, bones of 6-week old mice were examined. microCT data showed that cKO mice had less trabeculae in distal femur, and stereological analysis revealed that bone volume and mineral density in cKO mice were significantly decreased, manifested as lower trabecular number, thickness, and increased trabecular spacing (Figure 3A and Appendix A). To see dynamic actions of MACF1 on bone formation, mice were double labeled with calcein, and bone histomorphometric analysis revealed that matrix mineralization and bone formation in femur were significantly slower (Figure 3B).

Osteoblasts and osteoclasts are two main cell types in mature bone tissue that synthesize or decompose bone matrix to coordinate bone remodeling. To further understand the behavior of MACF1 in bone development, histochemistry approaches were utilized. To differentiate mineralized bone from osteoid during rapid bone growth, undecalcified femur sections (plastic sections) from 6-week old male mice were prepared. Goldner’s trichrome staining showed that in addition to porotic and lesser trabeculae, the cKO mice also had atypical accumulated osteoid on surfaces of both the trabecular and cortical bone (Figure 4A). Serological factors regarding bone formation and bone resorption were then analyzed, ELISA results showed that bone formation-related factors such as osteocalcin (OCN) were decreased, while bone resorption-related factors like carboxy-terminal collagen crosslinks (CTX) level were increased (Figure 4B). Further, femoral FFPE sections were prepared and TRAP staining showed that TRAP positive area was increased in cKO mice (Figure 4C). These data suggest that mesenchymal loss of MACF1 imbalanced osteoblastic and osteoclastic functions, and impaired mineralization capability on bone surfaces.

### 3.6. Loss of Microtubule Actin Crosslinking Factor 1 (MACF1) Weakens Bone Properties in Adult Mice

Bone mass peaks at the age of around 7-months in mice, and then decreases gradually with age. To confirm age-related phenotypes due to MACF1 deletion, mice of different ages were examined. At 3-months old, mineral density in cKO mice was lower as examined by DEXA, especially in hindlimb and lumbar vertebra (Figure 5A and Appendix A). Calcein double labelling showed that mineral apposition rate and bone formation rate in trabecular and cortical bone of cKO mice were both decreased significantly (Figure 5B). In addition, the three-point bending test showed in the cKO group not only reduced maximum compression stress and compression strain, but also reduced Young’s elastic modulus (Figure 5C), suggesting weakened toughness and strength of the bone in cKO mice.

At 7-months of age, microCT analysis showed that, in distal femur of cKO mice, bone mass and bone volume were significantly lower than those in Flox mice, lesser trabecular were found in cKO mice, and the trabecular microarchitecture deteriorated as well (Figure 5D). By 12-months of age, trabecular bone volume and thickness of cKO mice was even lower (Figure 5E), and, expectedly, medullary adipocytes in cKO mice increased in both number and size. It is worth noting that, with increased age, BV/TV (bone volume fraction) disparity between Flox and cKO mice increased from 23.9% at 6 weeks to 35.7% at 12 month (Figure 3A and Figure 5D), suggesting that the MACF1 deletion phenotype could last through adulthood, and loss of MACF1 impairs bone properties more seriously in aged mice. In summary, these results suggest that MACF1 positively regulates bone mass and bone quality in mice, and loss of MACF1 confers risk for degenerative phenotypes such as osteoporosis and fracture.

### 3.7. Microtubule Actin Crosslinking Factor 1 (MACF1) Interacts with SMAD Family Member 7 (SMAD7) in Mesenchymal Stem Cells (MSCs)

To explore novel downstream molecules that interact with MACF1 and regulate MSCs’ osteogenic differentiation, we performed an iTRAQ-labeled proteomics experiment using MC3T3-E1 preosteoblasts. First, we co-immunoprecipitated cell lysate using anti-MACF1 antibody, and confirmed immunoprecipitation capability of the anti-MACF1 antibody (Figure 6A and Appendix A). The IP products were then labeled with iTRAQ reagent and subjected to liquid chromatography followed by tandem mass spectrometry (LC-MS/MS) analysis. Unique peptide analysis showed that there were more than 60 proteins interacting with MACF1 in MC3T3-E1 cells, among which five were transcription factors (TFs), i.e., BHLHE40, HES1, REL, SMAD7, ZFP748 (Figure 6B). Functional similarity prediction using the ToppGene database showed that SMAD7 had the most potential to be involved in osteogenic functions (data for ZFP748 not found) (Figure 6C and Appendix A). In addition, qPCR analysis revealed that these TFs were all downregulated in *Macf1* knockdown cells, and the level of *Bhlhe40* and *Smad7* were relatively high in MC3T3-E1 cells (Figure 6D). Lastly, we verified SMAD7-MACF1 interaction in wild type MSCs. Co-IP result showed that SMAD7 was detectable in the anti-MACF1 immunoprecipitated products, indicating that MACF1 and SMAD7 interact with each other in MSCs (Figure 6E). Immunofluorescence showed from the perspective of intracellular distribution co-localization of MACF1 and SMAD7 (Figure 6F), which was consistent with the co-IP result.

On the basis of MACF1-SMAD7 interaction, to further clarify their regulatory pattern during osteogenic differentiation, we manipulated MACF1 expression in three cell lines/strains, and tested SMAD7 expressions in cytoplasm and nucleus, respectively. Firstly, in MSCs isolated from Flox mice, we found that MACF1 was mostly expressed in the cytoplasm, with a few in the nucleus; while in MSCs from cKO mice, expression of MACF1 decreased significantly in both the cytoplasm and nucleus. Similar to MACF1’s expression pattern, SMAD7 also concentrated mostly in the cytoplasm, and deletion of MACF1 significantly decreased expression of SMAD7, especially in the nucleus (Figure 6G and Appendix A). Next, the *Macf1* knockdown (MACF1-shRNA) MC3T3-E1 preosteoblasts were assayed, and we found that MACF1 and SMAD7 were both reduced in KD cells as compared with NC cells, the reduction was especially in obvious in the nucleus (Figure 6G and Appendix A). Lastly, we overexpressed MACF1 in KD cells by transfection of the pKH3S-ACF7 plasmid, and found that restoration of MACF1 significantly elevated the expression of SMAD7, not only in the cytoplasm, but also in the nucleus. These data suggest that during MSCs’ osteogenic differentiation, MACF1 interacts with SMAD7 and shares a similar expression trend.

### 3.8. Microtubule Actin Crosslinking Factor 1 (MACF1) Facilitates Nucleus Translocation of SMAD Family Member 7 (SMAD7) to Drive Osteogenic Differentiation

In the canonical TGF-β and BMP pathways in bone, cytoplasmic SMAD family member 7 (SMAD7) acts as a suppressor in the cytoplasm [37], but the nuclear roles of SMAD7 have not been fully clarified. To clarify how SMAD7 participates in osteogenic regulation through interacting with MACF1, we constructed a conventional SMAD7 overexpression plasmid (pSMAD7), a SMAD7 overexpression plasmid with mutant nuclear localization signal (SMAD7_NLS^Mut^), and a SMAD7 overexpression plasmid with normal nuclear localization signal (SMAD7_NLS), respectively (Figure 7A). To study the osteogenic functions of SMAD7, these plasmids were transfected into MC3T3-E1 osteoblasts.

Immunofluorescence images showed that 12 h after transfection, the pSMAD7 and SMAD7_NLS^Mut^ plasmid were both distributed mostly within the cytoplasm, while the SMAD7_NLS plasmid concentrated mostly in the nucleus (Appendix A). qPCR analysis revealed that, 5 days after osteo-induction, *Smad7* expression increased dramatically after cells were transfected with three kinds of plasmids (Figure 7B). In cells transfected with pSMAD7 plasmid, expressions of osteogenic markers and osteogenic-related transcription factors decreased significantly compared with empty vector transfected group (Figure 7C), in cells transfected with pSMAD7_NLS^Mut^ plasmid, changes of osteogenic markers and osteogenic-related TFs were similar with the pSMAD7 group. However, in cells transfected with NLS-carrying SMAD7 plasmid, levels of both osteogenic markers and osteogenic-related TFs increased significantly, even higher than that in the untransfected group (Figure 7C).

At last, transfected cells were stained. Alkaline phosphatase (ALP) staining showed that, as compared with the empty vector group, ALP activity decreased in both pSMAD7 and pSMAD7_NLS^Mut^ transfected cells, but increased dramatically in NLS-carrying plasmid transfected cells (Figure 7D). These results suggest that during osteogenic differentiation, cytoplasmic SMAD7 plays inhibitory roles, while the nuclear part positively regulates osteogenic differentiation. Concerning the direct interaction of MACF1 and SMAD7, these data indicate that during MSCs’ osteogenic differentiation, MACF1 facilitates nuclear translocation of SMAD7 to drive downstream osteo-related signaling to positively regulate bone formation.

## 4. Discussion

In this study, we identified MACF1 as a key regulator for bone development and formation. We show that MACF1 is necessary for maintaining osteogenic differentiation in MSCs. Loss of MACF1 decreases bone mass and quality in adult mice, which confers risk for degenerative bone diseases such as osteoporosis and fracture. Further, we show that a transcriptional factor SMAD7 interacts directly with MACF1, and initiates and promotes downstream osteogenic pathways when it has been translocated into the nucleus by MACF1. Our findings provide a new mechanism for understanding MSCs’ osteogenic differentiation and the regulation of bone formation, and highlight MACF1 as a potential therapeutic target for degenerative bone diseases such as osteoporosis.

### 4.1. Microtubule Actin Crosslinking Factor 1 (MACF1) Is Essential for Regulating Bone Development and Formation

Conventional or nervous-tissue-specific knockout of MACF1 was reported to be embryonic lethal [6,31], but mesenchymal-specific MACF1 KO mice survived and are physically normal, suggesting a relatively independent role of MACF1 in bones. However, mice lacking MACF1 show different degrees of development retardation in skull and long bones. Compared with those deficiency phenotypes found in other MACF1 KO models [1,9,10,31,38,39], we believe that MACF1 is also responsible for positive regulation of bone growth and development.

These results are improvements of our previously published data. In our previous studies, the roles of MACF1 in osteoblastic mineralization was only investigated by in vitro knockdown techniques, and then the relationship between MACF1 and bone formation was further investigated by subcutaneous injection of MACF1 overexpression plasmid in the skull. Although the close relationship between MACF1 and bone formation has indeed been discovered, the in vivo implication of MACF1 in bone formation has not been depicted. In in vivo studies, the Sp7-Cre generated MACF1 cKO mice showed significant development retardation in the skull and long bones [40], but compared with Prx1-Cre generated MACF1 cKO mice, Sp7-Cre induced MACF1 deletion shows a greater inhibiting effect on long bones, while Prx1-Cre induced MACF1 deletion shows a greater inhibiting effect on the skull. Concerning the promoting effect of foreign MACF1 plasmid on skull development, these data indicate adequate capability of mesenchymal MACF1 in skull development. In addition, dental abnormalities were common with Sp7-Cre generated, but not Prx1-Cre generated, MACF1 cKO mice, indicating a subtle difference of MACF1 in bone forming lineage cells, and also diverse roles of MACF1 at different developmental stages. In summary, we propose that MACF1 is essential for bone formation.

Interestingly, mesenchymal deletion of MACF1 elevated bone surface adipocyte numbers and osteoclastic activities were not mentioned in previous studies. Many factors are responsible for committed differentiation of MSCs. In MSCs of MACF1 cKO mice, osteoblastic differentiation is suppressed, while adipogenic differentiation is promoted, especially in aged mice, this could be an immediate cause for adipocyte accumulation. Bone modeling occurs at early developmental stage, in which the formation modeling and resorption modeling are relatively independent, and contribute to bone mass much faster; while bone remodeling converts woven bone to lamellar bone, and the osteoblastic function is closely related with osteoclastic function. MACF1 deficiency induced bone remodeling uncoupling could be an indirect cause for elevated osteoclastic activity.

### 4.2. Microtubule Actin Crosslinking Factor 1 (MACF1) Interacts Directly with SMAD Family Member 7 (SMAD7) to Initiate Osteogenic Differentiation

We identify SMAD7 as a novel downstream target of MACF1. In the canonical TGF-β/BMP signaling, SMAD7 serves as an inhibitory SMAD to inhibit bone formation in a negative feedback way [37,41]. In this study, we found distinct roles of SMAD7 in cytoplasm and nucleus of MSCs. The cytoplasm SMAD7 still plays inhibitory roles as reported in canonical TGF-β/BMP signaling, while the nucleus SMAD7 is found to positively regulate osteogenic differentiation. Previously, the positive role of SMAD7 was only found in cells such as osteoblasts [42] and myoblasts [43]. Our study is the first mention of SMAD7’s subcellular roles in osteogenic differentiation, which confirms that SMAD7 could positively regulate bone formation.

Further, we show that MACF1 could mediate nuclear translocation of SMAD7, and MACF1 directly targets SMAD7 to promote bone formation. MACF1 is reported to have ATPase activity [5], and mediate intracellular material trafficking [6,44]. Additionally, there is data to show that SMAD7 has multiple phosphorylation sites [45]. Previously we found that in osteoblasts, MACF1 could shuttle between the cytoplasm and nucleus; in Wnt/β-catenin and BMP2 signaling, MACF1 was shown to play vital roles, especially in β-catenin nuclear translocation. By combining results obtained in this study, we propose that SMAD7 is a downstream target of MACF1. However, how MACF1 facilitates SMAD7 nuclear translocation, and the role of MACF1 in the canonical TGF-β/BMP signaling remains to be investigated.

### 4.3. Microtubule Actin Crosslinking Factor 1 (MACF1) Is a Novel Potential Target for Primary Osteoporosis

Through expression profiling and tissue PCR verification, we discovered that the mesenchymal MACF1 level is significantly decreased in clinical osteoporotic patients. MACF1 is responsible for multiple key processes such as cellular integrity, cell migration, and differentiation. In many studies, age-related loss of MACF1 is associated with degenerative changes to a varying degree. Neuronal deletion of MACF1 impairs proliferation and differentiation capability [1,9,16], and is likely to cause Parkinson’s disease among the elderly [14]. Loss of MACF1 also exacerbates pressure overload induced left ventricular (LV) hypertrophy, and leads to LV dilation and contractile dysfunction [38].

As a large structural protein, we believe that age-related loss of MACF1 is common in multiple tissues. In bone cells, MACF1 and cytoskeleton give cells shape and integrity, and thus form the rigid bone tissue. During the aging process, bones become osteoporotic and fragile, and mesenchymal MACF1’s downregulation in osteoporotic patients suggest that, in the progression of primary osteoporosis, functional deficiency of MACF1 might be a key cause. This proposes MACF1 as a key potential target for primary osteoporosis in the perspective of molecular medicine.

In summary, we discovered new roles of MACF1 in regulating osteogenic differentiation and bone formation, and that loss of MACF1 in bone forming cells impairs osteogenic functions and then leads to degenerative bone disorders such as osteoporosis. Further, we identified SMAD7 as a key transcriptional factor that interacts with MACF1 to be translocated into the nucleus to initiate downstream osteogenic pathways.

## Figures and Tables

**Figure 1 cells-09-00616-f001:**
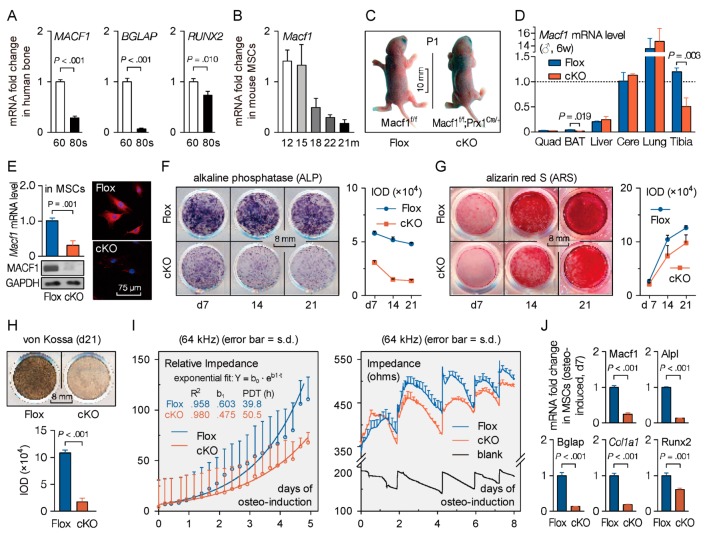
Mesenchymal deletion of microtubule actin crosslinking factor 1 (MACF1) attenuates osteogenic differentiation capability. (**A**) Relative expression level of *MACF1* and osteogenic genes in bone specimens. Postmenopausal patients in their 60s or 80s were compared. (**B**) Relative expression level of *Macf1* in mesenchymal stem cells (MSCs) isolated from different ages of mice. (**C**) Representative images of neonatal MACF1 conditional knockout (cKO) mice (P1, postnatal day 1). (**D**) Real-time quantitative PCR analysis of *Macf1* mRNA levels in organs or tissues of MACF1 cKO mice (6-week old, male). Quad, quadriceps. BAT—brown adipose tissue; Cere—cerebrum. (**E**) Expression and distribution of MACF1 in cKO MSCs. *Gapdh*/GAPDH were used as internal references. (**F**,**G**) Representative staining images showing alkaline phosphatase expression and mineralized nodules formation in osteo-induced cKO MSCs. Staining intensity was quantified as IOD (integrated optical density). (**H**) Representative images of von Kossa staining showing mineralization capability in osteo-induced cKO MSCs. (**I**) Real-time impedance curve showing proliferation (left box) and differentiation capability (right box) of osteo-induced cKO MSCs. PDT—population doubling time. (**J**) Real-time PCR analysis of osteogenic marker genes in osteo-induced cKO MSCs (d7). Data are represented as mean ± s.d. Gene expression was normalized by *GAPDH*/*Gapdh*. Significances were determined using Student’s *t*-test.

**Figure 2 cells-09-00616-f002:**
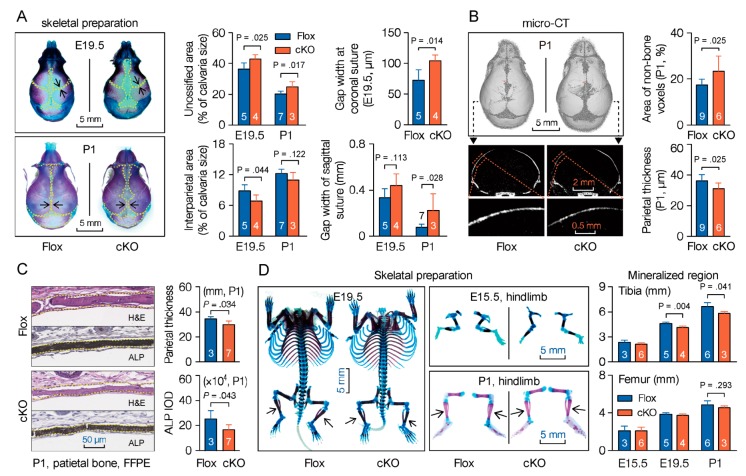
Mesenchymal deletion of MACF1 retards early stage bone development in mice. (**A**) Representative images of cleared skeletal preparation showing early-stage development of the skull (E19.5—embryonic day 19.5; P1—postnatal day 1). Red shows ossified tissue, while blue indicates cartilage (dashed lines). Gap width indicates the nearest distance between adjacent bones. (**B**) Representative 3D reconstructed images showing skull development at P1. Cross-section representation of the parietal bone is also shown. Non-bone voxels indicate an unossified part. (**C**) Representative images of hematoxylin and eosin (H&E) and alkaline phosphatase (ALP) staining showing morphology and ALP expression in parietal bone (P1, coronal sections). (**D**) Representative images of cleared skeletal preparation showing ossification in long bones. Arabic numeral at the bottom of each bar indicates the number of mice used in the experiments. Data are represented as mean ± s.d. Significances were determined using Student’s *t*-test.

**Figure 3 cells-09-00616-f003:**
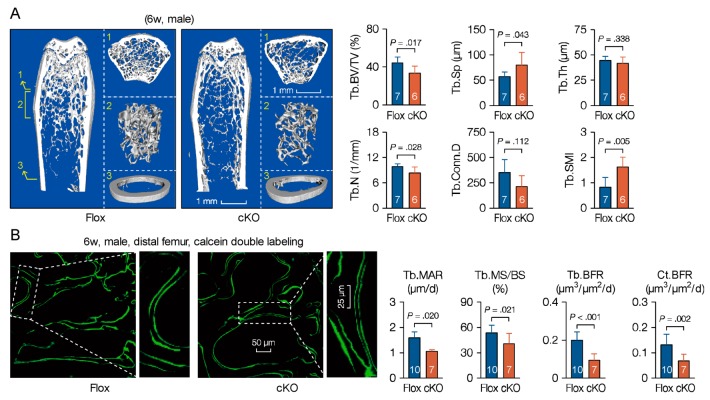
Weakened bone properties and bone formation capability in MACF1 cKO mice. (**A**) Representative 3D reconstructed images showing microarchitecture of the distal femur (6-week old, male). Positions of reconstructed regions are indicated by Arabic numerals. Stereological parameters for trabecular bone in distal femur. BV/TV—bone volume fraction; Tb.N—trabecular number; Tb.Sp—trabecular spacing; Tb.Th—trabecular thickness; Conn.D—connectivity density; SMI—structural model index. (**B**) Representative images of calcein double labeling showing mineral apposition and bone formation in the femur (male, 6-week old, coronal sections). Boxed regions are enlarged. MAR—mineral apposition rate; MS/BS—mineralizing surface per bone surface; BFR—bone formation rate per bone surface. Arabic numeral at the bottom of each bar indicates the number of mice used in the experiments. Data are represented as mean ± s.d. Significances were determined using Student’s *t*-test.

**Figure 4 cells-09-00616-f004:**
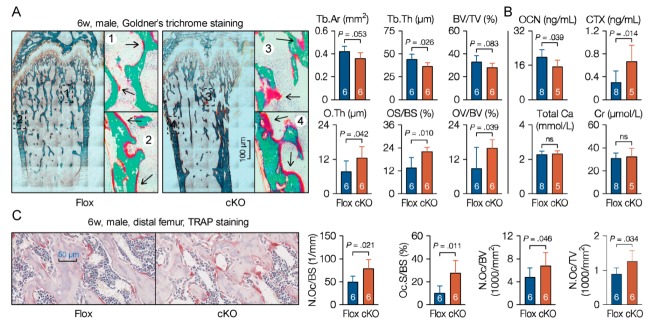
Reduced osteoblastic activity on bone surfaces of MACF1 cKO mice. (**A**) Representative images of Goldner’s trichrome staining showing mineralized matrix and osteoid in the femur (6-week old, male, coronal sections). Green shows mineralized bone, while red indicates osteoid. Numbered boxes are enlarged to show details. OV/BV—osteoid volume; OS/BS—osteoid surfaces; O.Th—average osteoid thickness. (**B**) Sandwich enzyme immunoassay and biochemical assay for in vitro quantification of serological factors. OCN—osteocalcin; CTX—carboxy-terminal collagen crosslinks; Ca—calcium; Cr—creatinine. (**C**) Representative images of TRAP staining showing bone resorption near proximal metaphysis (6-week old, male, coronal sections). N.Oc—osteoclasts numbers; Oc.S—osteoclast surfaces. Arabic numeral at the bottom of each bar indicates the number of mice used in the experiments. Data are represented as mean ± s.d. Statistical significance was determined using Student’s *t*-test.

**Figure 5 cells-09-00616-f005:**
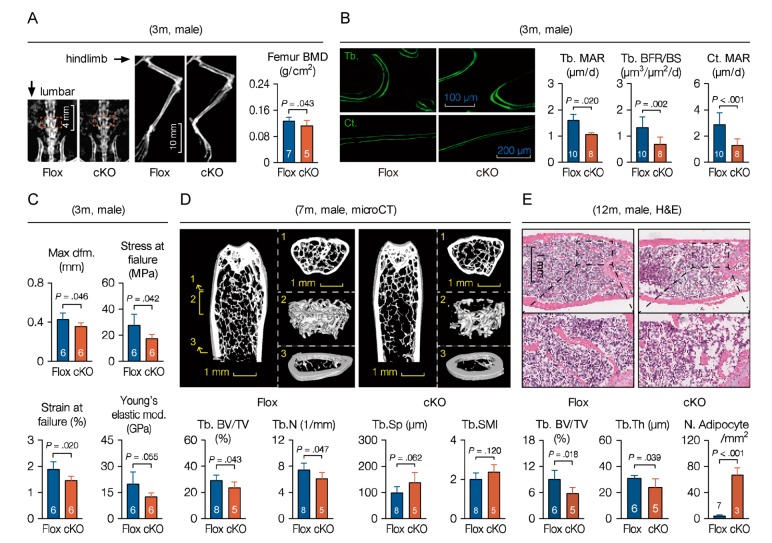
Weakened bone properties and bone formation capability in adult and aged cKO mice. (**A**) X-ray images of the MACF1 cKO mice (3-month old, male). (**B**) Representative images of calcein double labeling showing mineral apposition and bone formation in distal femur (3-month old, male, coronal sections). Tb.—trabecular; Ct.—cortical. (**C**) Femoral mechanical property (male, 3-month old). Max dfm.—maximum deformation; Young’s mod.—Young’s elastic modulus. (**D**) Representative 3D reconstructed images showing microarchitecture of distal femur (7-month old, male). Positions of reconstructed regions are indicated by Arabic numerals. Stereological parameters for trabecular bone in distal femur were quantified. (**E**) Representative images of femoral H&E staining (12-month old, male, FFPE, coronal sections). Arabic numeral at the bottom of each bar indicates the number of mice used in the experiments. Data are represented as mean ± s.d. Statistical significance was determined using Student’s *t*-test.

**Figure 6 cells-09-00616-f006:**
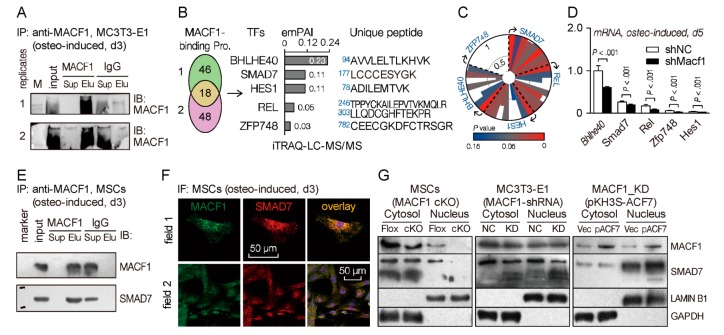
MACF1 interacts directly with SMAD7. (**A**) Co-immunoprecipitated products for iTRAQ analysis. Sup—supernatant (unbound protein); Elu—eluate (Pro-Ab-Beads complex). (**B**) MACF1-interacting proteins identified by LC-MS/MS. emPAI—exponentially modified protein abundance index. (**C**) Potentials of candidate TFs to be involved in osteogenic differentiation. The radial coordinate represents the *Score* value predicted by ToppGene database, while the angular coordinate shows eight predicted items (clockwise: GO: Molecular Function; GO: Biological Process; GO: Cellular Component; Human Phenotype; Mouse Phenotype; Pathway; Pubmed; Disease). (**D**) Real-time PCR analysis of candidate TFs expression levels in MACF1 knockdown MC3T3-E1 preosteoblasts (osteo-induced, d5). (**E**) Co-IP assay showing interaction of MACF1 and SMAD7 in osteo-induced wildtype MSCs. (**F**) Expression and distribution of MACF1 and SMAD7 in wildtype MSCs (osteo-induced, d3). (**G**) Western blot analysis of MACF1 and SMAD7 levels in MACF1 cKO MSCs, MACF1 knockdown (KD) preosteoblasts, and MACF1-stable-overexpressed KD cells. ACF7 (actin cross-linking factor 7) is a synonym for MACF1. Lamin B1 and GAPDH were used as internal reference for nucleus and cytoplasm, respectively. Data are represented as mean ± s.d. Statistical significance was determined using Student’s *t*-test.

**Figure 7 cells-09-00616-f007:**
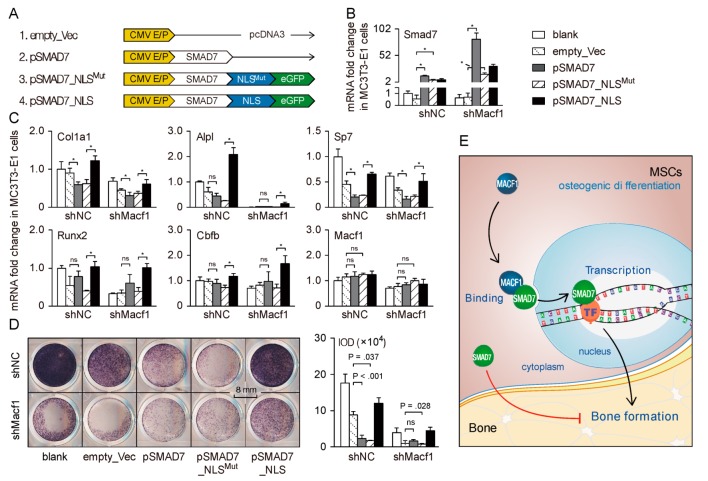
MACF1 facilitates SMAD7 nucleus translocation to drive osteogenic differentiation. (**A**) Schematic diagram of different SMAD7 overexpression plasmids. CMV E/P—CMV enhancer and promoter; NLS—nuclear localization sequence; eGFP—enhanced green fluorescent protein. (**B**,**C**) Real-time PCR analysis of *Smad7* and osteogenic-related genes in MC3T3-E1 preosteoblasts 5 days after plasmid transfection. (**D**) Representative staining images showing alkaline phosphatase activity in MC3T3-E1 preosteoblasts 5 days after plasmid transfection. (**E**) Working model. MACF1 is essential for maintaining MSCs’ osteogenic differentiation and bone formation. In MSCs, cytoplasmic SMAD7 inhibits osteogenic differentiation, while MACF1 promotes nuclear translocation of SMAD7 to initiate osteogenic pathways. Data are represented as mean ± s.d. Statistical significance were determined using Student’s *t*-test, and *p*-value less than 0.05 was considered statistically significant (marked by a asterisk).

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
