# Peer review of "Mesenchymal MACF1 Facilitates SMAD7 Nuclear Translocation to Drive Bone Formation"

_cells, 2020, doi:10.3390/cells9030616_

Round 1
Reviewer 1 Report
In general, this is an excellent manuscript that sheds considerable light on an intriguing and important finding. The study is well designed and methodologically carried out. The conclusions are well supported, particularly as this relates to the regulation of bone formation. Publication is recommended.
Author Response
Feb 26, 2020
Dear Reviewer,
Thanks for the quick reply. We have carefully read your comments on the manuscript entitled “Mesenchymal MACF1 Facilitates SMAD7 Nuclear Translocation to Drive Bone Formation”. We really appreciate the effort you made for reviewing this manuscript, your kind comments on this are very much appreciated. Thank you again and look forward to having your updated opinions on this manuscript.
Sincerely,
Airong Qian (on behalf of all the authors)
Ph.D., Professor of School of Life Sciences
Northwestern Polytechnical University
No. 127, West Youyi Rd., Xi'an, Shaanxi, 710072, China
qianair@nwpu.edu.cn, Tel/Fax: +86-29-88491840 (office). Mobile: +86 13572108260
Reviewer 2 Report
The work describes the role of MACF1 during osteogenic differentiation. They also provide insights into the mechanisms that might underlay this regulation. SMAD7 seems to be a good target in this study.
Then authors used a variety of biological resources in their investigations, going from mouse models, primary cells, to cell lines. The techniques used in this study were appropriate to address the questions raised.
Major comments:
The main finding in this study is that MACF1 is a key regulator of the osteogenic differentiation process. However, as recognized by the authors, its precise role in the TGFb signaling pathway remains elucive.
In addition, the nature of interactions between MACF1 and SMAD7 is not described neither discussed in this paper. It is known that SMAD7 can be phosphorylated. Is the translocation process from the cytoplasm to the nucleus dependent of SMAD7 phosphorylation?
Osteogenic differentiation process and ECM secretion involves RUNX2 activation through SMAD phosphorylation. In a situation where SMAD7 is translocated into the nucleus inducing a positive signal for the osteogenic differentiation, what is the contribution of the other SMAD to this process? These questions were not raised neither experimentally, nor in the discussion to weigh the contribution of this pathway to the whole osteogenic process. For instance, are the other pathways silenced? Is SMAD1 phosphorylation level reduced when SMAD7 is translocated into the nucleus?
Minor comments:
Line 321: the authors show that cKO cells produced slightly less ECM than controls.
Line 329: they show that cKO cells proliferate less and differentiate less than controls.
Could the decrease in ECM production be due to the presence of less cKO cells than controls, producing hence less ECM?
How did the authors account for this knowing that ECM production in cKO cells was only slightly reduced?
Line 144: replace were analysis with were analyzed
Lines 396 and 397: change bone formation and formation with bone formation and bone resorption
Line424: replace foung by found
Define BV/TV
Author Response
Feb 26, 2020
Dear Reviewer,
Thanks for the quick reply. We have carefully read your comments on the manuscript entitled “Mesenchymal MACF1 Facilitates SMAD7 Nuclear Translocation to Drive Bone Formation”. We really appreciate the effort you made for reviewing this manuscript, your kind comments on this are very much appreciated.
Based on your comments, after discussed with main authors of this manuscript, we have now completed the revision of this manuscript. The responses to your comments are listed below in the Q-A format. We hope that all these corrections and revisions would be satisfactory:
Q1: "The main finding in this study is that MACF1 is a key regulator of the osteogenic differentiation process. However, as recognized by the authors, its precise role in the TGFb signaling pathway remains elucive".
A: Thanks, this is indeed an important point, and the evidence we provided in the submitted manuscript is not strong enough to clarify MACF1’s precise role in the TGF-β pathway. It is almost certain, however, that both MACF1 and SMAD7 are involved in the TGF-β pathway. This study focuses mainly on MACF1's role in primary osteoporosis, SMAD7 is an extension of MACF1's mechanism into primary osteoporosis. Since the TGF-β pathway is fundamental to osteogenesis, to further understand their precise roles in the TGF-β pathway, we think it’s better to start a new project to fully discuss the roles of MACF1 and SMAD7 in TGF-β signaling.
Q2: "In addition, the nature of interactions between MACF1 and SMAD7 is not described neither discussed in this paper. It is known that SMAD7 can be phosphorylated. Is the translocation process from the cytoplasm to the nucleus dependent of SMAD7 phosphorylation? "
A: We are sorry that the interactions between MACF1 and SMAD7 was not fully discussed. MACF1 is reported to have ATPase activity (Wu, X. et al. Cell 2008, 135, 137–148), and can mediate intracellular material trafficking (Chen, H.-J. et al. Genes Dev. 2006, 20, 1933–45. Slep, K.C. et al. J. Cell Biol. 2005, 168, 587–598); Also there are data to show that SMAD7 has multiple phosphorylation sites (Hornbeck, P. V. et al. Nucleic Acids Res. 2015, 43, D512–D520, including potential phosphorylation sites predicted by database). Previously we found that in osteoblasts, MACF1 could shuttle between the cytoplasm and nucleus; In Wnt/β-catenin and BMP2 signaling, MACF1 was shown to play vital roles, especially in β-catenin nuclear translocation. So phosphorylation-dependent SMAD7 nuclear translocation in MSCs is still a hypothesis based on the evidence we have provided. In addition, in the mentioned new project, phosphorylation-mediated SMAD7 nuclear translocation will be tested, and the roles of other SMAD proteins will be intensively studied.
In the revised version of the manuscript, we have made some changes to this part, and highlighted the interaction between MACF1 and SMAD7 (see DISCUSSION 4.2 in the revised manuscript).
Q3: “Osteogenic differentiation process and ECM secretion involves RUNX2 activation through SMAD phosphorylation. In a situation where SMAD7 is translocated into the nucleus inducing a positive signal for the osteogenic differentiation, what is the contribution of the other SMAD to this process? These questions were not raised neither experimentally, nor in the discussion to weigh the contribution of this pathway to the whole osteogenic process. For instance, are the other pathways silenced? Is SMAD1 phosphorylation level reduced when SMAD7 is translocated into the nucleus”
A: As of now, we still don’t have enough data to show the role of other SMADs during SMAD7 nuclear translocation, this is important but still complicated.
In the canonical BMP pathway in bones, SMAD7 inhibits SMAD1/5/8 nuclear translocation by inhibiting their phosphorylation, and its inhibitory role is mainly found in the cytoplasm, the nuclear translocation action of SMAD7 is not well recognized (Luo K. CSH Perspect Biol, 2017, 9(1): a022137; Yoon J H, et al. Nat Commun, 2015, 6(1): 1-14). In the N-end, R-SMADs and Co-SMAD share the homologous MH1 domain, which is reported to participate in nuclear import (Chapter 17, Signal Transduction (Third Edition), 2016); However, MH1 domain is not found in SMAD7, this raise a hypothesis that other SMADs may be involved in SMAD7 nuclear translocation.
So we think a new project would be necessary to fully address this issue. In this project, the involvement of SMAD7 in the TGF-β and BMP signaling pathway would be investigated systematically, including other potentially silenced pathways and SMAD1 phosphorylation.
Based on these, we have added new content to the Discussion part of the revised manuscript to further discuss the involvement of other components in TGF-β and BMP signaling with SMAD7.
Minor comments:
Q4: Line 321: the authors show that cKO cells produced slightly less ECM than controls.
A: We don't have data to show ECM in MSCs. Do you mean the ECIS curve (Figure 1I, and text in line 331)? The ECIS system uses cell-substrate impedance values to show differentiation capability of cultured cells, higher values indicate higher capability during induced differentiation. Detailed procedure can be found in the Material & Method part (see M&M 2.16).
Q5: Line 329: they show that cKO cells proliferate less and differentiate less than controls. Could the decrease in ECM production be due to the presence of less cKO cells than controls, producing hence less ECM? How did the authors account for this knowing that ECM production in cKO cells was only slightly reduced?
A: As in Q4, you may refer to ECIS. Our data showed that, compared with Flox cells (MSCs), the cKO cells showed less potential to proliferate and differentiate. MACF1 is a large structural protein that could interact simultaneously with both F-actin cytoskeleton and microtutule cytoskeleton, and cytoskeleton give shape to cells and provide tracks for intracellular material transport. Less potential in cKO cells is likely to be a reflect of such roles.
Q6: Line 144: replace were analysis with were analyzed
A: Revised. See line 145 in the revised manuscript.
Q7: Lines 396 and 397: change bone formation and formation with bone formation and bone resorption
A: Pending. Figure 3B shows the formation modeling process, no resorption modeling data were provided.
Q8: Line424: replace foung by found
A: Revised. See line 435 in the revised manuscript
Q9: Define BV/TV
A: Revised. See line 439 in the revised manuscript
Thank you again and look forward to having your updated opinions on this manuscript.
Sincerely,
Airong Qian (on behalf of all the authors)
Ph.D., Professor of School of Life Sciences
Northwestern Polytechnical University
No. 127, West Youyi Rd., Xi'an, Shaanxi, 710072, China
qianair@nwpu.edu.cn, Tel/Fax: +86-29-88491840 (office). Mobile: +86 13572108260
Reviewer 3 Report
The manuscript by Zhao et al expands previous findings by the group into the function of MACF1 in bone development. Mice with a knockdown of MACF1 in osteochondro progenitor cells were created with Prx1-Cre and demonstrate mild skeletal defects in the long bones but larger effects in the cranium, here especially in the suture areas. These findings are in line with published in vitro data that shows delayed osteogenic differentiation of MSc from MACF1-KO mice or cell lines with a MACF1 deletion and are identical to what has recently been described with a MACF1 osterix-cre knockdown strategy. Furthermore, the manuscript quantifies MACF1 gene expression in mice and human mesenchymal stem cells and shows an age dependent decline in expression. This observation is however not further explained by the results or mentioned in the discussion.
MACF1 has already been shown to facilitate nuclear translocation of beta catenin through its interaction with axin to enhance axin binding to LRP6 upon Wnt stimulation. Furthermore, the Qian group has just published that MACF1 knockdown attenuates Bmp2/Smad/Runx2 signalling. The manuscript expands these findings to include MACF1's interaction with SMAD7, which is a negative regulator of TGFbeta signalling. The authors show that in osteogenic cells, MACF1 can be found in the nuclear fraction together with SMAD7. However, the authors do not investigate the effect of stimulating the cells with TGFbeta, which in other cell types induces translocation of SMAD7 from the nucleus to the cytoplasm. In general, the data on the localisation of SMAD7 and MACF1 is not as clear as the rest of the results. In figure 6F, image quality and size need to be improved as currently it is impossible to see the localisation of the molecules. Figure 6D should be quantified as the quality of the Western blots is low. For the SMAD7 transfection results in figure 7B and C clearer colour discrimination between the different conditions is needed - both bars of pSMAD7_NLSmut and normal have the same shade.
Detailed comments:
More information needs to be added to the figure legends, especially in figure 2, 3 and 5 where bars contain numbers which are not explained. The numbers of independent experiments/mice should be stated both in the methods and in the figure legends.
More information on the knockdown of MACF1 in MC3T3 cells needs to be added. The method section should contain a brief statement on how (by which method) these cells were obtained.
Line 29: ‘MACF1 expression’
Line 128: Type II collagenase
Line 170: (Leica TCS SP8, Germany)
Line 396 and 397, change one bone formation to bone resorption.
Line 533, delete 'novel'
Lines 581-586: overinterpretation of results.
In general, all abbreviations should be introduced when they are first used. Ensure that chemical symbols are correctly displayed by using lower case numbers.
Author Response
Feb 26, 2020
Dear Reviewer,
Thanks for the quick reply. We have carefully read your comments on the manuscript entitled “Mesenchymal MACF1 Facilitates SMAD7 Nuclear Translocation to Drive Bone Formation”. We really appreciate the effort you made for reviewing this manuscript, your kind comments on this are very much appreciated.
Based on your comments, after discussed with main authors of this manuscript, we have now completed the revision of this manuscript. The responses to your comments are listed below in the Q-A format. We hope that all these corrections and revisions would be satisfactory:
Q1: "Furthermore, the manuscript quantifies MACF1 gene expression in mice and human mesenchymal stem cells and shows an age dependent decline in expression. This observation is however not further explained by the results or mentioned in the discussion"
A: Thanks. As one of the most fundamental research background, this means a lot to this paper. In the revised manuscript, we have added necessary explanation to the DISCUSSION part to stress the age-dependent decline of MACF1 (see DISCUSSION part 4.3), and the added content would be clearer to link MACF1 to age-dependent bone formation reduction.
Q2: "The authors show that in osteogenic cells, MACF1 can be found in the nuclear fraction together with SMAD7. However, the authors do not investigate the effect of stimulating the cells with TGFbeta, which in other cell types induces translocation of SMAD7 from the nucleus to the cytoplasm".
A: Thanks, this is also important, and the evidence we provided in the submitted manuscript is not strong enough to clarify MACF1’s precise role in the TGF-β pathway. It is almost certain, however, that both MACF1 and SMAD7 are involved in the TGF-β pathway. This study focuses mainly on MACF1's role in primary osteoporosis, SMAD7 is an extension of MACF1's mechanism into primary osteoporosis. Since the TGF-β pathway is fundamental to osteogenesis, to further understand their precise roles in the TGF-β pathway, we think it’s better we start a new project to fully discuss the role of MACF1 and SMAD7 in TGF-β signaling. In this project, the involvement of SMAD7 in the TGF-β and BMP signaling pathway would be investigated systematically, phosphorylation-mediated SMAD7 nuclear translocation will be tested, and the roles of other SMAD proteins will be intensively studied.
Q3: "In general, the data on the localization of SMAD7 and MACF1 is not as clear as the rest of the results. In figure 6F, image quality and size need to be improved as currently it is impossible to see the localisation of the molecules. Figure 6D should be quantified as the quality of the Western blots is low ".
A: The original figures submitted to the editorial office are in high resolution, with details that show clearly cell borders and colored fluorescence. We may contact with the editorial office later.
Q4: "For the SMAD7 transfection results in figure 7B and C clearer colour discrimination between the different conditions is needed - both bars of pSMAD7_NLSmut and normal have the same shade".
A: When preparing figures for this manuscript, grayscale bars were preferred, but they appeared blurred in low resolution. We have changed the appearance of these histograms, and now these bars are easier to be distinguished from each other.
Q5: "More information needs to be added to the figure legends, especially in figure 2, 3 and 5 where bars contain numbers which are not explained. The numbers of independent experiments/mice should be stated both in the methods and in the figure legends".
A: These numbers are mice numbers. We have included these informations in figure legends where necessary. We didn’t include such information in the Method part because certain method (most of which are common techniques) may be used at different time point, and at each time point, mice numbers are not always the same, to do so may increase reading obstacle.
Q6: "More information on the knockdown of MACF1 in MC3T3 cells needs to be added. The method section should contain a brief statement on how (by which method) these cells were obtained".
A: Technical procedure of how MACF1 knockdown MC3T3-E1 cell line was established has been reported in a paper published by our group (Lifang Hu, Airong Qian, et al. BMB Rep. 2015, 48(10):583-8.), and we have cited it in the manuscript (citation 20).
Other comments:
Line 29: ‘MACF1 expression’. A: Revised.
Line 128: Type II collagenase. A: Revised.
Line 170: (Leica TCS SP8, Germany). A: Revised.
Line 396 and 397, change one bone formation to bone resorption. A: Pending. Figure 3B shows the formation modeling process, no resorption modeling data were provided.
Line 533, delete 'novel'. A: Revised.
Lines 581-586: overinterpretation of results. A: Revised.
All abbreviations should be introduced when they are first used. A: The full text has been searched and revised.
Spelling of chemical symbols. A: Symbols in the original manuscript submitted to the editorial office was correctly displayed, we have also corrected all these misspellings in the revised manuscript.
Thank you again and look forward to having your updated opinions on this manuscript.
Sincerely,
Airong Qian (on behalf of all the authors)
Ph.D., Professor of School of Life Sciences
Northwestern Polytechnical University
No. 127, West Youyi Rd., Xi'an, Shaanxi, 710072, China
qianair@nwpu.edu.cn, Tel/Fax: +86-29-88491840 (office). Mobile: +86 13572108260
Reviewer 4 Report
The authors conducted in vivo and in vitro studies to reveal effects of MACF1 on osteogenesis. They showed that MACF1 deficient mouse showed impaired skeletal development in bone morphometry analysis. They also identified SMAD7 as a MACF1 direct target. MACF1 facilitated SMAD7 nuclear translocation which underlies MACF1-mediated osteogenesis. Overall, the study was well designed and the results were clear. This study is well matched to the scope of Cells and worth publishing in this journal. Only a few minor issues need to be addressed before the publication.
Line 342, “paired related homeobox1” should be written when Prx1 was first described in the manuscript.
Lines 363 to 368, “Unlike flat bones …” should be in main text, not in figure 2 legend.
Author Response
Feb 26, 2020
Dear Reviewer,
Thanks for the quick reply. We have carefully read your comments on the manuscript entitled “Mesenchymal MACF1 Facilitates SMAD7 Nuclear Translocation to Drive Bone Formation”. We really appreciate the effort you made for reviewing this manuscript, your kind comments on this are very much appreciated.
Based on your comments, after discussed with main authors of this manuscript, we have now completed the revision of this manuscript. The responses to your comments are listed below in the Q-A format. We hope that all these corrections and revisions would be satisfactory:
Q1: Line 342, “paired related homeobox1” should be written when Prx1 was first described in the manuscript.
A: Revised. We also searched and added full name to other abbreviations when first mentioned.
Q2: Lines 363 to 368, “Unlike flat bones …” should be in main text, not in figure 2 legend.
A: This is a typographical error, we have made revision in the main text.
Thank you again and look forward to having your updated opinions on this manuscript.
Sincerely,
Airong Qian (on behalf of all the authors)
Ph.D., Professor of School of Life Sciences
Northwestern Polytechnical University
No. 127, West Youyi Rd., Xi'an, Shaanxi, 710072, China
qianair@nwpu.edu.cn, Tel/Fax: +86-29-88491840 (office). Mobile: +86 13572108260
Round 2
Reviewer 3 Report
Thank you for considering my comments in the resubmission. To clarify, in lane 396/397 (now 417/418) the sentence states that 'serological factors regarding bone formation and bone formation were analyzed'. It should read: 'seroloigal factors regarding bone formation and bone resorption were analyzed' as this is what is presented in figure 4B.
I am still concerned about the size of Figure 6F and the quality of figure 6G.
The complexity of the figures and size of subfigures could be addressed by reducing the data on the MACF1 knowdown animals as these are almost identical to the MACF1 knockdown using SP7, which has recently been published. This would enable highlighting the novel data such as reduction of MACF1 expression with age and the interaction with SMAD7.
Author Response
Feb 27, 2020
Dear Reviewer,
Thanks for the quick reply. We have carefully read your comments on the manuscript entitled “Mesenchymal MACF1 Facilitates SMAD7 Nuclear Translocation to Drive Bone Formation”. Thank you for your sustained attention on this manuscript.
Based on your comments, after discussed with main authors of this manuscript, we have now completed the revision of this manuscript. The responses to your comments are listed below in the Q-A format.
Q1: "To clarify, in lane 396/397 (now 417/418) the sentence states that 'serological factors regarding bone formation and bone formation were analyzed'. It should read: 'seroloigal factors regarding bone formation and bone resorption were analyzed' as this is what is presented in figure 4B".
A: Revised. We are so sorry that this error was not corrected, thanks for pointing this out again.
Q2: "I am still concerned about the size of Figure 6F and the quality of figure 6G".
A: We have resized panel Figure 6F, it should now appear clearer.
We are sorry that we did not elaborate Figure 6G’s issue in the previous response. Before starting WB experiment, we firstly tested the specificity and sensitivity of the antibodies used. There are several commercial suppliers that provide MACF1 antibody, the MACF1 Rabbit pAb was purchased from Abcam (ab117418), we have been using this antibody for WB for many years, its performance was stable, and WB band quality was satisfactory. But the situation for SMAD7 antibody was a little bit different, commercially available SMAD7 antibodies can only be found at ProteinTech (i.e. SMAD7 Rabbit pAb 25840-1-AP and SMAD7 Mouse mAb 66478-1-Ig). We then requested for trial packs, and tested those using MSCs and MC3T3-E1 cells. The test showed that the mouse mAb was less sensitive than rabbit pAb. Therefore, the rabbit pAb (25840-1-AP) was preferred for formal experiments. Although the SMAD7 rabbit pAb meets the requirements of the experiment, there's still a stability concern. In some cases, the SMAD7 bands were not as clean and clear as usual. So we repeated this WB experiment for several times, and provided the data in the supplementary material (Figure S5E) for additional support.
Q3: "The complexity of the figures and size of subfigures could be addressed by reducing the data on the MACF1 knowdown animals as these are almost identical to the MACF1 knockdown using SP7, which has recently been published. This would enable highlighting the novel data such as reduction of MACF1 expression with age and the interaction with SMAD7".
A: Thanks. This is an important point, and we have also discussed this issue previously. We have recently published a MACF1 cKO mouse model (using the Sp7-Cre), compare with the Prx1-Cre model in this paper, the former deletes Macf1 gene in osteoblasts, while the latter inactivates Macf1 gene in mesenchymal stem cells. We did considered to combine these two project into one, but after discussed with main authors of both project, we agreed to keep the two projects independent, this is due to their difference in the biological processes involved, and the difference in bone phenotype. In this two models, although the phenotype defect in bones are both evident, specific situation is not the same. As mentioned in the Discussion part, the Sp7-cre induced cKO mice show more significant defect in long bones than in the skull, the Prx1-cre induced cKO is however the opposite. We attribute this difference to the specific roles of MACF1 in different bone forming lineage cells. Since MACF1's in vivo roles in bone formation is not well studied, we think it’s necessary to retain these data to make the readers aware of this difference. In future studies, we will continue to improve the knowledge of MACF1 in bone formation.
Thank you again and look forward to having your updated opinions on this manuscript.
Sincerely,
Airong Qian (on behalf of all the authors)
Ph.D., Professor of School of Life Sciences
Northwestern Polytechnical University
No. 127, West Youyi Rd., Xi'an, Shaanxi, 710072, China
qianair@nwpu.edu.cn, Tel/Fax: +86-29-88491840 (office). Mobile: +86 13572108260